# Simplified Approach to Nonlinear Vibration Analysis of Variable Stiffness Plates

**Jorge Andérez González and Riccardo Vescovini \***

Dipartimento di Scienze e Tecnologie Aerospaziali, Politecnico di Milano, Via La Masa 34, 20156 Milano, Italy
*   Correspondence: riccardo.vescovini@polimi.it; Tel.: +39-02-2399-8332

**Abstract:** A formulation for the analysis of the nonlinear vibrations of Variable Stiffness (VS) plates is presented. Third-order Shear Deformation Theory (TSDT) is employed in conjunction with a mixed variational formulation. The solution is sought via Ritz approximation for the spatial dependency, while time dependency is handled via Differential Quadrature (DQ) and Harmonic Balance (HB) methods. The main advantage of the framework is the reduced computational time, which is of particular interest to explore the large design space offered by variable stiffness configurations. The results are validated against reference solutions from the literature. Exemplary parametric studies are presented to demonstrate the potential of the approach as a powerful means for exploring the nonlinear vibration response of VS plates.

**Keywords:** nonlinear vibrations; variable stiffness plates; semi-analytical methods

## 1. Introduction

The possibility of tailoring the internal load paths via variable stiffness (VS) designs is a topic of increasing interest in the field of modern composite structures. These new designs are achieved by allowing the fibers to run along curvilinear trajectories, thus enlarging the design space with respect to classical straight-fiber composites.

The advantages of this design approach have been discussed in previous efforts in the literature. The thermal and mechanical buckling behavior is assessed in [1–3], while the nonlinear postbuckling response is investigated in [4,5]. Linear free vibrations have been studied too, see [6], in the framework of a variable-kinematic approach. On the contrary, a relative lack of studies is found in the field of large-amplitude vibrations, which are a topic of crucial importance for many engineering applications, of which aerospace panels are an example. Indeed, these structures are commonly forced to vibrate at a large amplitude by their operating environment and so could benefit by an improved VS tailoring. Potential advantages have been discussed in [7], where the authors developed a finite element approach with higher-order elements in conjunction with FSDT. The study concluded that the possibility of steering fibers allows plates to behave in a more rigid fashion with respect to straight-fiber configurations. Despite the above mentioned potential, the field of nonlinear vibration of VS plates appears to have been just partially explored in the literature.

As a matter of fact, most of the works in the literature refer to the analysis of isotropic and straight-fiber composites [8–11]. Early works rely on an analytical approach to the subject, where classical strategies based on elliptic functions and the Galerkin and Navier methods were proposed to handle the spatial part [12–14]. In the referenced works, the time dependency is accounted for via perturbation approaches. Despite the inherent efficiency of these analytical strategies, their field of employ is restricted to specific boundary conditions and constitutive laws. The finite element method has been proposed as an excellent means for extending the field of employ, an early application of which is found for isotropic plates in [15]. Within the FE framework, different ways for handling time dependency are found,

such as direct integration [16] and the Harmonic Balance (HB) method [17]. Interesting studies combining Hierarchical finite elements and HB method are found in [18,19], where the convergence of the solution is studied and the stability assessed via Floquet theory.

In the field of VS panels, the literature on nonlinear vibrations is still relatively scarce and, in most cases, refers to FE-based approaches to discretize the spatial dependency. Pioneering work in the field is due to Ribeiro and co-workers [7,20]. In these efforts, the p- and h-version of the finite element method are used to study the forced oscillations and steady-state free vibrations via the Newmark and HB methods, respectively. The excellent review of [21] clarifies the potential of VS designs to shape vibration modes and frequencies and, at the same time, the need for optimization based-approaches to handle the increased design complexity. Another application of finite elements to the nonlinear vibrations of VS plates is found in [22], where high-order FEM is employed in conjunction with HB and classical lamination theory. The study is restricted to fully clamped plates, but interestingly demonstrates that the highest and lowest nonlinear frequencies are achieved via VS configuration.

Despite the above-mentioned advantages offered by a FEM-based approach, it is believed to be of interest to explore alternative strategies that can promote computational saving. This aspect is particularly relevant for VS configurations, as the possibility of steering the fibers largely increases the number of the design variables. In this regard, semi-analytical strategies are a natural candidate, as they have been employed with success to the nonlinear analysis of isotropic and straight-fiber configurations [23,24], but have been rarely extended to VS configurations [25].

In a previous effort by one of the authors, a simplified semi-analytical approach has been presented with a focus on thin VS plates [25]. However, the literature is relatively scarce when extending the field of application to moderately thick plates. Among the few works, Refs. [26,27] should be mentioned, where the nonlinear thermoelastic dynamics is analyzed in referring to Third-order Shear Deformation Theory (TSDT).

A new framework is proposed here to address the nonlinear vibrations of VS plates based on TSDT. The formulation presented here appears to be the first attempt to combine a higher-order theory and a mixed approach in the context of nonlinear vibration analysis. Solution strategies are developed by combining the Ritz method along with the Differential Quadrature (DQ) and the Harmonic Balance (HB) methods. A simplified, single-mode approach is then proposed, such that preliminary assessments can be performed on the fly. The new mixed variational framework developed here is not restricted to nonlinear vibration problems, but can be applied to solve other nonlinear problems in the fields of nonlinear bending and postbuckling responses.

The paper is organized as follows: Section 2 illustrates the theoretical framework by presenting the mixed variational principle at the basis of the proposed approach; Section 3 introduces the approximations for the spatial and the time dependency; Section 4 aims at illustrating the comparison against results from the literature and between different strategies for handling the time dependence; then, the potential application to perform a parametric study is presented.

## 2. Formulation

A formulation is developed for assessing the nonlinear dynamic response of laminated plates with VS properties. A sketch of the structure is reported in Figure 1. The plate volume is defined as $\Omega \equiv A \times [-h/2 \leq x_3 \leq h/2]$, where $A$ is the plate midsurface and $h$ denotes the thickness. A Cartesian coordinate system is taken over the plate midsurface, where the axes $x_1$ and $x_2$ run along the two planar directions; the corresponding dimensions are denoted with $a$ and $b$, respectively. The $x_3$ axis is taken according to the right-hand rule and runs along the thickness direction. In view of future developments, it is useful to introduce the nondimensional coordinates $\xi = 2\dfrac{x_1}{a}$ and $\eta = 2\dfrac{x_2}{b}$, such that $(\xi, \eta) \in [-1\ 1]$. The structure is laminated with a symmetric stacking sequence, where each ply has constant density $\rho$, over the domain, and is assumed to be perfectly bonded to the other plies.

The fibers are allowed to exhibit a non-constant orientation along the in-plane directions. Specifically, the orientation is assumed to vary linearly between the center and the edge of the plate. The two orientations at these locations are denoted with $T_0$ and $T_1$, as depicted in Figure 1.

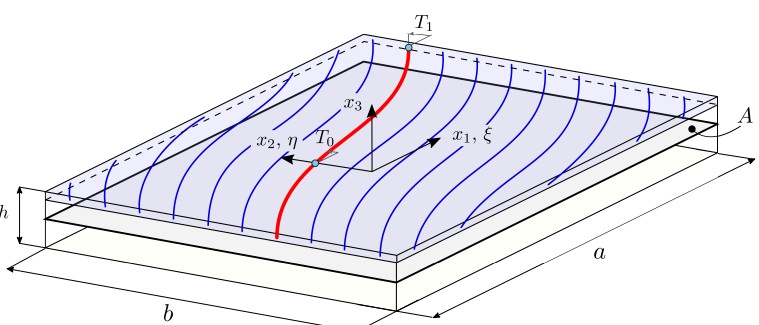

**Figure 1.** VS plate: reference system and fiber path.

The orientation of each single ply is denoted here with the compact notation $\langle T_0|T_1 \rangle$, where the following distribution is implied [1]:

$$\theta = 2\frac{T_1 - T_0}{a}|x_1| + T_0 \tag{1}$$

The linear distribution allows the fiber path to be defined by means of two values only. Hence, parametric studies can be performed with relative ease, making straightforward graphical representations for the response of interest possible. More complex fiber distributions can be specified, see, e.g., [28]. In general, they can be beneficial in further extending the design space, at the cost of a larger number of variables to be handled.

The study of nonlinear vibrations is carried out by considering an external load in the form of a pressure $p = p(x_\alpha)$ directed along the axis $x_3$. The case of uniform distribution can be considered as a special case, as well as a concentrated force being retrieved by introducing a Dirac delta distribution.

### 2.1. Kinematics and Generalized Strains

The underlying kinematic model relies upon TSDT, offering the advantage of allowing the study of thin and relatively thick plates, with no need for a shear factor to be defined. This model is well-known in the literature, see, e.g., [29,30], and its features are not reported in detail for the sake of brevity. Within the present approach, the effect of normal stretching is not accounted for.

The in-plane directions $x_\alpha$ are denoted through the index $\alpha = 1, 2$, while the out-of-plane direction is identified by the index 3. Accordingly, the components of the displacement field can be split into $U_\alpha$ and $U_3$. The third-order kinematic model reads:

$$\begin{aligned}
U_\alpha(x_\alpha, x_3, t) &= u_\alpha(x_\alpha, t) + x_3\varphi_\alpha(x_\alpha, t) + c_1 x_3^3[\varphi_\alpha(x_\alpha, t) + u_{3,\alpha}(x_\alpha, t)] \\
U_3(x_\alpha, x_3, t) &= u_3(x_\alpha, t)
\end{aligned} \tag{2}$$

where the expansion is carried out up to the third-order in $x_3$. The generalized displacements $u_\alpha$, $u_3$ $\varphi_\alpha$ are a function of time and the planar coordinates $x_\alpha$ only; in addition, $c_1 = -\dfrac{4}{3h^2}$ is a constant allowing the Cauchy equilibrium to be satisfied at the top and the bottom of the laminate, as shown later.

A modified version of the Green–Lagrange strain tensor is introduced as:

$$2\epsilon_{ik} = U_{i,k} + U_{k,i} + U_{3,i}U_{3,k} \tag{3}$$

where the nonlinear terms are restricted to the out-of-plane displacements, consistently with the von Kármán nonlinear plate theory; a comma followed by an index denotes differentiation with respect to that index. Upon substitution of Equation (2) into Equation (3), one obtains:

$$\epsilon_{\alpha\beta} = \xi_{\alpha\beta} + x_3 k_{\alpha\beta} + c_1 x_3^3 \hat{k}_{\alpha\beta}, \qquad \gamma_{\alpha3} = \left(1 + 3c_1 x_3^2\right)\zeta_\alpha \qquad (4)$$

where the strain measures are:

$$2\xi_{\alpha\beta} = u_{\alpha,\beta} + u_{\beta,\alpha} + u_{3,\alpha}u_{3,\beta}, \qquad 2k_{\alpha\beta} = \varphi_{\alpha,\beta} + \varphi_{\beta,\alpha}, \qquad 2\hat{k}_{\alpha\beta} = \varphi_{\alpha,\beta} + \varphi_{\beta,\alpha} + 2u_{3,\alpha\beta}$$
$$\zeta_\alpha = \varphi_\alpha + u_{3,\alpha} \qquad (5)$$

The terms in Equation (5) are the generalized strain parameters of the kinematic model and are expressed as a function of the unknown displacements $u_\alpha$, $u_3$ and $\varphi_\alpha$.

### 2.2. Generalized Forces

The internal stresses are available from the strains by application of the constitutive law. Under the assumption of transversally isotropic material, the stress–strain relation is expressed as:

$$\sigma_{\alpha\beta} = \overline{Q}^{\text{p}}_{\alpha\beta\eta\omega}(x_\alpha)\epsilon_{\eta\omega}, \qquad \sigma_{\alpha3} = \overline{Q}^{\text{t}}_{\alpha3\eta3}(x_\alpha)\gamma_{\eta3} \qquad (6)$$

where $\overline{Q}^{\text{p}}_{\alpha\beta\eta\omega}$ and $\overline{Q}^{\text{t}}_{\alpha\beta\eta\omega}$ denote the elastic tensors for the plane stress and transverse shear components, the overline indicating that the coefficients are expressed in the laminate system $x_1 - x_2$. Furthermore, the dependency of the coefficients on the planar coordinates $x_\alpha$ is reported explicitly to give evidence of a peculiar feature of non-straight fiber configurations. By inspection of Equations (4) and (6), it is immediate to verify that $\sigma_{\alpha3}(x_\alpha, \pm h/2, t) = 0$, i.e., the Cauchy equilibrium is identically satisfied at the top and the bottom of the laminate.

Within the kinematic model illustrated earlier, the following generalized forces are found to be energetically conjugated with the strain measures of Equation (5):

$$N_{\alpha\beta} = \langle\sigma_{\alpha\beta}\rangle = A_{\alpha\beta\eta\omega}\xi_{\eta\omega}, \qquad\qquad M_{\alpha\beta} = \langle x_3\sigma_{\alpha\beta}\rangle = D_{\alpha\beta\eta\omega}k_{\eta\omega} + F_{\alpha\beta\eta\omega}\hat{k}_{\eta\omega}$$
$$P_{\alpha\beta} = \left\langle c_1 x_3^3 \sigma_{\alpha\beta}\right\rangle = F_{\alpha\beta\eta\omega}k_{\eta\omega} + H_{\alpha\beta\eta\omega}\hat{k}_{\eta\omega}, \qquad Q_\alpha = \left\langle \left(1 + 3c_1 x_3^2\right)\sigma_{\alpha3}\right\rangle = A^*_{\alpha3\eta3}\zeta_\eta \qquad (7)$$

where the symbol $\langle\cdot\rangle = \sum_k \int_{h_k}^{h_{k+1}} \cdot \, dx_3$ is introduced to denote the integral over the plies along the thickness-wise direction. The elastic coefficients appearing in Equation (7) are obtained as:

$$\{A_{\alpha\beta\eta\omega}, D_{\alpha\beta\eta\omega}, F_{\alpha\beta\eta\omega}, H_{\alpha\beta\eta\omega}\} = \left\langle \{1, x_3^2, c_1 x_3^4, c_1 x_3^6\}\overline{Q}^{\text{p}}_{\alpha\beta\eta\omega}\right\rangle$$
$$A^*_{\alpha3\eta3} = \left\langle \left(1 + 3c_1 x_3^2\right)^2 \overline{Q}^{\text{t}}_{\alpha3\eta3}\right\rangle \qquad (8)$$

Note that the laminate constitutive relations of Equation (7) are obtained under the assumption of symmetric layup, so any integral of odd functions in $x_3$ vanishes, providing no contribution to laminate constitutive law.

### 2.3. Variational Framework

The approach developed here relies upon a weak-form formulation of the problem, which proved to be effective in the application of direct solution methods [25,28,31,32]. The underlying variational principle is an extension of the early work due to Giavotto [33], originally developed for thin-plates, and extended here to the case of third-order theory. An application of the strong-form formulation of an analogous mixed strategy is found in [34] for the postbuckling analysis of transversally isotropic plates. The formulation is mixed inasmuch the in-plane displacements $u_\alpha$ are removed from the set of unknowns

and are replaced with a potential function of the membrane resultants, i.e., the Airy stress function $F \equiv F(x_\alpha)$. In particular, this function allows membrane forces to be derived as:

$$N_{\alpha\beta} = e^{\alpha\eta} e^{\beta\omega} F_{,\eta\omega} \tag{9}$$

where $e^{\alpha\eta}$ is the 2D permutation symbol. Based on Equation (9), the in-plane strains can be expressed as a function of $F$ upon inversion of the first of Equation (7):

$$\xi_{\alpha\beta} = a_{\alpha\beta\eta\omega} N_{\eta\omega} = a_{\alpha\beta\eta\omega} e^{\alpha\eta} e^{\beta\omega} F_{,\eta\omega} \tag{10}$$

where $a_{\alpha\beta\eta\omega} = A^{-1}_{\alpha\beta\eta\omega}$ is the fourth order tensor expressing the membrane compliance.

The main steps to derive the mixed variational principle are outlined below. The starting point is the Hamilton's principle, whose expression reads:

$$\delta \int_{t_1}^{t_2} \Pi \, dt = 0 \tag{11}$$

with:

$$\Pi = K - (U + V) \tag{12}$$

where $K$ and $U$ are the kinetic and strain energy, respectively, while $V$ is the potential of the external loads. Owing to the kinematic model of Equation (2) and assuming that in-plane inertia is negligible, the kinetic energy is expressed as:

$$\begin{aligned} K &= \frac{1}{2} \int_\Omega \rho \left( \dot{U}_\alpha^2 + \dot{U}_3^2 \right) d\Omega = \\ &\approx \frac{1}{2} \int_A \left( I_0 \dot{u}_3^2 + I_1 \dot{\varphi}_\alpha^2 + I_2 \dot{u}_{3,\alpha}^2 + 2 I_3 \dot{\varphi}_\alpha \dot{u}_{3,\alpha} \right) dA \end{aligned} \tag{13}$$

where odd integrals of $x_3$ are zero as the density is assumed to be constant all over the plate. The inertial terms $I_k$ are obtained upon integration along the thickness of the relevant terms appearing in Equation (13), and they are obtained as:

$$I_0 = \rho h, \qquad I_1 = \frac{17}{315} \rho h^3, \qquad I_2 = \frac{1}{252} \rho h^3, \qquad I_3 = -\frac{4}{315} \rho h^3 \tag{14}$$

The strain energy is readily available by recalling Equations (5) and (7):

$$\begin{aligned} U &= \frac{1}{2} \int_\Omega \left( \sigma_{\alpha\beta} \epsilon_{\alpha\beta} + \sigma_{\alpha 3} \gamma_{\alpha 3} \right) d\Omega = \\ &= \frac{1}{2} \int_A \left( \xi_{\alpha\beta} N_{\alpha\beta} + k_{\alpha\beta} M_{\alpha\beta} + \hat{k}_{\alpha\beta} P_{\alpha\beta} + \zeta_\alpha Q_\alpha \right) dA \end{aligned} \tag{15}$$

The potential of the external loads, in the presence of a pressure acting along the vertical direction, is:

$$V = -\int_A u_3 p \, dA \tag{16}$$

The variational principle of Equation (11) refers to the functional $\Pi \equiv \Pi(u_\alpha, u_3, \varphi_\alpha)$, which can be used within the framework of a displacement-based approach. A mixed approach is pursued here, so the expression of the relevant functional requires some further elaboration. Specifically, an augmented functional is introduced, where the in-plane strains $\xi_{\alpha\beta}$ are taken as additional unknowns of the problem. Accordingly, the in-plane strain components must be subjected to a compatibility condition that is added to Equation (12) to achieve an augmented functional, whose expression reads:

$$\begin{aligned} \Pi_{\text{aug}} &\equiv \Pi_{\text{aug}} \left( u_\alpha, u_3, \varphi_\alpha, \lambda_{\alpha\beta}, \xi_{\alpha\beta} \right) = \\ &= \Pi + \int_A \lambda_{\alpha\beta} \left( \xi_{\alpha\beta} - \frac{1}{2} u_{\alpha,\beta} - \frac{1}{2} u_{\beta,\alpha} - \frac{1}{2} u_{3,\alpha} u_{3,\beta} \right) dA \end{aligned} \tag{17}$$

where $\lambda_{\alpha\beta}$ are the Lagrange multipliers enforcing the in-plane compatibility requirements. By application of the techniques of the calculus of variations, one can easily find the equations expressing the stationarity condition of Equation (17), i.e., the Euler–Lagrange equations of the problem:

$$
\begin{cases}
\lambda_{\alpha\beta} - N_{\alpha\beta} = 0 \\
\xi_{\alpha\beta} - \frac{1}{2}u_{\alpha,\beta} - \frac{1}{2}u_{\beta,\alpha} - \frac{1}{2}u_{3,\alpha}u_{3,\beta} = 0 \\
\lambda_{\alpha\beta,\beta} = 0 \\
Q_{\alpha,\alpha} - P_{\alpha\beta,\alpha\beta} + u_{3,\alpha\beta}\lambda_{\alpha\beta} + u_{3,\beta}\lambda_{\alpha\beta,\alpha} = I_0\ddot{u}_3 - I_2\ddot{u}_{3,\alpha\alpha} - I_3\ddot{\varphi}_{\alpha,\alpha} - p \\
M_{\alpha\beta,\alpha\beta} + P_{\alpha\beta,\beta} - Q_\alpha = I_1\ddot{\varphi}_\alpha + I_3\ddot{u}_{3,\alpha}
\end{cases}
\tag{18}
$$

The first condition of Equation (18) demonstrates that the Lagrange multipliers can be understood as the membrane resultants—this is expected from an energy interpretation of Equation (17). Accordingly, the third stationary condition provides the membrane equilibrium condition, which is identically satisfied owing to Equation (9). The second condition is the constraint equation imposed via the Lagrange multiplier technique, while the last two equations provide the dynamic equilibrium in the vertical direction and the rotational ones, respectively.

A new functional can be derived by substituting back the first of Equation (18) into Equation (17) and recalling Equations (9) and (10). The following expression is so obtained:

$$
\begin{aligned}
\Pi^* \equiv \Pi^*(u_3, \varphi_\alpha, F) = \\
= K - (U^* + V)
\end{aligned}
\tag{19}
$$

which is the functional at the basis of the proposed variational approach that replaces $\Pi$ in Equation (11), and where:

$$
U^* = \frac{1}{2}\int_A \left( -\xi_{\alpha\beta}N_{\alpha\beta} + k_{\alpha\beta}M_{\alpha\beta} + \hat{k}_{\alpha\beta}P_{\alpha\beta} + \zeta_\alpha Q_\alpha \right) \mathrm{d}A + \frac{1}{2}\int_A N_{\alpha\beta}u_{3,\alpha}u_{3,\beta}\,\mathrm{d}A
\tag{20}
$$

which represent the sum of bending and shear energies with the membrane complementary one.

The functional $\Pi^*$ depends on the unknowns $u_3$, $\varphi_\alpha$ and $F$ only; any dependence on the in-plane displacement components $u_\alpha$ is removed. This is clearly seen from the expressions of the kinetic energy $K$, Equation (13), the potential of external loads $V$, Equation (16), and the partially inverted strain energy, Equation (20), in combination with Equations (7) and (10).

This result can be viewed as a generalization of the variational approaches of [25,28,31,32,35] for thin plates to incorporate transverse shear deformability in the framework of third-order plate theory. Note that the application proposed in this work regards the nonlinear vibrations, but the same variational principle can be used for other nonlinear problems, such as the post-buckling one.

### 2.4. Boundary Conditions

The assumption of symmetrical layup has the effect of preventing any linear coupling between in-plane and out-of-plane responses; see Equation (7). However, a nonlinear coupling exists between membrane and flexural responses in the context of the von Kármán large deflection theory, as seen from the first of Equation (5). Hence, boundary conditions need to be specified in terms of flexural and membrane behavior. The former conditions are those associated with the unknowns $u_3$ and $\varphi_\alpha$ (essential) and their energy conjugated counterparts $M_{\alpha\beta}$, $P_{\alpha\beta}$ and $Q_{\alpha\beta}$ (natural). These conditions can be free (F), simply-supported (S) and clamped (C) edges, according to their standard definition [29]. Capital letters denote the edge constraint and are reported in sequence, starting from the one at $\xi = -1$ and proceeding in a counterclockwise direction. In-plane boundary conditions involve the in-plane displacements $u_\alpha$ and the membrane forces, both expressed in terms of the Airy stress function $F$. The following conditions are defined: unsupported (U), held (H) and

immovable (I) edge. In the first case, the edge is completely free to translate; in the second one, the edge is kept straight, while allowed to translate; in the third one, the in-plane displacement along the normal direction is prevented. The in-plane shear vanishes in all the three cases, as tangential displacements are free. A sketch of the conditions is reported in Figure 2.

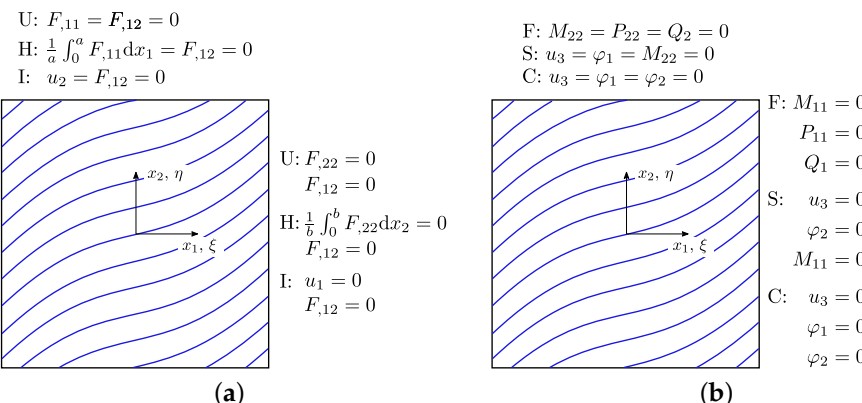

**Figure 2.** Comparison between different methods: (**a**) in-plane conditions; (**b**) flexural conditions.

## 3. Approximate Solution

The mixed variational formulation presented earlier is well suited to be applied in the context of direct methods. Specifically, the Ritz method is employed here for handling the spatial dependency; the resulting ordinary differential equations (ODEs) in the time variable are then solved using different techniques: the Differential Quadrature method and the Arc-length Harmonic Balance method, this latter in its multi- and single-mode formulations. The main features of both spatial and time approximation are outlined below.

### 3.1. Spatial Approximation via Ritz Method

The spatial dependency is approximated using global functions. This is an effective means for solving the problem with an excellent accuracy-to-number of degrees of freedom ratio. Within the framework of the Ritz method, the unknowns are expanded as:

$$
\begin{aligned}
\begin{Bmatrix} u_3(x_\alpha, t) \\ \varphi_x(x_\alpha, t) \\ \varphi_y(x_\alpha, t) \end{Bmatrix} &= \begin{bmatrix} \mathbf{N}^{\mathrm{w}}(x_\alpha) & & \\ & \mathbf{N}^{'x}(x_\alpha) & \\ & & \mathbf{N}^{'y}(x_\alpha) \end{bmatrix} \begin{Bmatrix} \hat{\mathbf{w}}(t) \\ \hat{\mathbf{\textit{x}}}(t) \\ \hat{\mathbf{\textit{y}}}(t) \end{Bmatrix} \\
&= \mathbf{N}^{\mathrm{u}}(x_\alpha)\hat{\mathbf{u}}(t) \\
F(x_\alpha, t) &= \mathbf{N}^{\mathrm{f}}(x_\alpha)\hat{\mathbf{\Phi}}(t)
\end{aligned}
\tag{21}
$$

where $\mathbf{N}^{\mathrm{u}}$ and $\mathbf{N}^{\mathrm{f}}$ are the matrices collecting the trial functions, obtained as the product between Legendre polynomials and boundary functions. The former were found to guarantee numerical robustness and fast convergence properties [36], the latter are introduced to enforce the essential boundary conditions. Note that, for the Airy stress function $F$, a split is operated between the internal and boundary terms: this strategy allows the approach to be formulated in a more convenient way, see [4]. The vectors $\hat{\mathbf{u}}$ and $\hat{\mathbf{\Phi}}$ of Equation (21) collect the unknown amplitudes associated with the Ritz expansion.

Upon substitution of Equation (21) into Equation (19) and by imposing the stationarity condition, one obtains the discrete equations governing the problem:

$$
\begin{cases}
\mathbf{M}\ddot{\hat{\mathbf{u}}} + \mathbf{K}\hat{\mathbf{u}} + \mathbf{n}(\hat{\mathbf{\Phi}}, \hat{\mathbf{w}}) = \mathbf{f} \\[2mm]
\mathbf{U}\hat{\mathbf{\Phi}} + \dfrac{1}{2}\hat{\mathbf{w}}^{\mathrm{T}} N_{mn} \hat{\mathbf{w}} = \mathbf{0}
\end{cases}
\tag{22}
$$

where **M**, **K**, and **U** are the mass, stiffness, and in-plane compliance matrices, respectively; the terms **n** and $N_{mn}$ are the vector and the matrix associated with the geometric nonlinearity; dot denotes derivative with respect to time. The first of Equation (22) defines the dynamic equilibrium, the second expresses the in-plane compatibility requirements. Note that the compatibility equation is time independent as the in-plane inertia is neglected. Hence, a static condensation can be operated upon substitution of the second equation into the first.

The governing equations can be further elaborated by expressing Equation (22) in modal coordinates, via transformation $\hat{\mathbf{u}} = \mathbf{V}\mathbf{q}$, where **V** is the matrix collecting the modes of vibration retained in the modal expansion, and **q** are the corresponding amplitudes. It follows that the final set of nonlinear ODEs in the time variable are obtained in the form:

$$\ddot{\mathbf{q}} + \mathbf{C}^{(q)}\dot{\mathbf{q}} + \mathbf{K}^{(q)}\mathbf{q} + \mathbf{n}^{(q)}(\mathbf{q}, \mathbf{q}, \mathbf{q}) = \mathbf{f}^{(q)} \tag{23}$$

where the superscript $(q)$ is introduced to identify all the terms obtained through projection onto the modal coordinates. It is worth noting that modal truncation allows the number of degrees of freedom to be reduced. Linear modes are assumed here to effectively capture the nonlinear response, which is generally acceptable unless the problem is strongly nonlinear.

Using an alternative indicial representation, which is useful for future developments, Equation (23) can be re-stated as:

$$\ddot{q}_k + \mathrm{diag}(2\zeta_k \omega_k)\dot{q}_k + \mathrm{diag}(\omega_k^2)q_k + \sum_{rst=1}^{n} \beta_{krst}^{(q)} q_r q_s q_t = f_k^{(q)} \tag{24}$$

where $\zeta_k$ is the modal damping ratio and $\beta_{krst}$ the coefficients of the nonlinear stiffness term.

### 3.2. Differential Quadrature Method

The Differential Quadrature method has been extensively used in the literature as a means for approximating the spatial dependency in differential problems; see, e.g., [32,37]. On the contrary, the DQ method is used here for handling time dependence and transforming the ODEs arising from the Ritz approximation into a set of nonlinear algebraic equations. The application of DQ for handling time dependency is relatively rare in the literature, but offers an intriguing approach for achieving a good balance between accuracy and computational time [38]. A time grid is firstly generated, where the time interval is divided into a number of $N_e$ intervals. Each interval is characterized by a number of sampling points denoted as $N$. Among the different distributions available, the Chebyshev one was found to guarantee improved accuracy when compared to others, such as the uniform distribution [38]. For this reason, the sampling points are taken as:

$$t_k = t_0 + \frac{1}{2}\left[1 - \cos\left(\frac{k-1}{N-1}\pi\right)\right]\left(t_f - t_0\right) \tag{25}$$

where the initial and final time are denoted as $t_0$ and $t_f$, respectively.

Within the framework of the DQ method, the time derivatives can be written as [39]:

$$\frac{\mathrm{d}^m \mathbf{q}}{\mathrm{d}t^m} = \mathcal{D}^{(m)}\mathbf{q} \tag{26}$$

where $\boldsymbol{\mathcal{D}}^{(m)}$ is a matrix of weighting coefficients expressing the order $m$ derivatives and given by:

$$
\boldsymbol{\mathcal{D}}^{(1)} = \begin{cases} \prod\limits_{i=1,i\neq k,j}^{N} (t_k - t_i) \bigg/ \prod\limits_{i=1,i\neq j}^{N} (t_j - t_i), & k \neq j \\ \sum\limits_{i=1,i\neq k}^{N} \dfrac{1}{t_k - t_i}, & k = j \end{cases}
$$

$$
\boldsymbol{\mathcal{D}}^{(m)} = \boldsymbol{\mathcal{D}}^{(1)} \boldsymbol{\mathcal{D}}^{(m-1)}
\tag{27}
$$

The expression above illustrates that large values of $N$ may lead to a progressively smaller denominator in Equation (27), with consequent effects on the stability of the numerical solution. For this reason, the time span is divided into a number of $N_e$ time elements, each one containing $N$ sampling points.

Upon substitution of Equation (26) into Equation (23), a nonlinear algebraic system is obtained as:

$$
[\mathbf{A} + \mathbf{A}_{\mathrm{nl}}(\mathbf{x}, \mathbf{x})]\mathbf{x} = \mathbf{b}
\tag{28}
$$

where the vector $\mathbf{x}$ is defined as $\mathbf{x} = \{q_{11} \ldots q_{nN}\}^{\mathrm{T}}$, and $n$ is the number of modes retained in the modal expansion. The two matrices appearing in Equation (28) are obtained as:

$$
\mathbf{A} = \left( \boldsymbol{\mathcal{D}}^{(2)} \otimes \mathbf{I} + \boldsymbol{\mathcal{D}}^{(1)} \otimes \mathbf{C}^{(q)} + \mathbf{I} \otimes \mathbf{K}^{(q)} \right)
$$

$$
\mathbf{A}_{\mathrm{nl}} = \mathbf{I} \otimes \mathbf{K}_{\mathrm{nl}}^{(q)}
\tag{29}
$$

Representing the expansion of the damping and stiffness matrices on the DQ sampling points via Kronecker multiplication; the matrix $\mathbf{I}$ is the identity of dimension $N$.

The problem of Equation (28) is then modified in order to account for the initial conditions. Specifically, the first and the last $n$ rows of $\mathbf{A}$—those associated with the first and the last sampling points, respectively—are substituted with the initial conditions on displacement and velocity.

The so-obtained nonlinear system is solved via Newton–Raphson iterations, which are continued up to satisfaction of a convergence criterion. The norm of the residual is considered for this scope.

### 3.3. Arc-Length Harmonic Balance Method

The Harmonic Balance method is a well-known approach for handling the temporal dependency in nonlinear vibration problems [40,41]. In the present framework, this approach offers the advantage of allowing both stable and unstable branches to be successfully captured. Furthermore, this method is a suitable starting point for deriving simplified analytical solutions, as discussed next.

According to the Harmonic Balance method, a truncated Fourier series is introduced for approximating the steady state solution as:

$$
\mathbf{q}(t) = \mathbf{Q}_0 + \sum_{n=1}^{H} \left[ \mathbf{Q}_{\mathrm{c}n} \cos(n\Omega t) + \mathbf{Q}_{\mathrm{s}n} \sin(n\Omega t) \right]
\tag{30}
$$

where $\mathbf{Q}_0$ is the amplitude of the constant contribution, while $\mathbf{Q}_{\mathrm{c}n}$ and $\mathbf{Q}_{\mathrm{s}n}$ are the amplitudes of the n-th cosine and sine term, respectively. In the present formulation, the expansion is carried out up to $H = 5$. While more terms could be retained, a number of preliminary tests revealed this truncation to be adequate in most cases. This choice is in agreement with previous works in the literature; see, e.g., [20,25]. The main idea of the

approach is the substitution of Equation (30) into Equation (24) followed by the collection of the corresponding harmonic terms. According to this procedure, one obtains:

$$\mathbf{r}(\mathbf{x}) \equiv \left[ -\Omega^2 \tilde{\mathbf{M}} + \Omega \tilde{\mathbf{C}} + \tilde{\mathbf{K}} + \tilde{\mathbf{n}}(\mathbf{x}, \mathbf{x}) \right] \mathbf{x} - \tilde{\mathbf{f}} = \mathbf{0} \tag{31}$$

where the expressions of $\tilde{\mathbf{M}}$, $\tilde{\mathbf{C}}$, $\tilde{\mathbf{K}}$ and $\tilde{\mathbf{n}}$ are relatively cumbersome and are not reported here for the sake of brevity; the vector $\mathbf{x}$ collects the unknown amplitudes of the expansion operated via Equation (30), i.e.,:

$$\mathbf{x}^{\mathrm{T}} = \{\mathbf{Q}_0^{\mathrm{T}} \ \mathbf{Q}_{c1}^{\mathrm{T}} \ \mathbf{Q}_{s1}^{\mathrm{T}} \ \cdots \ \mathbf{Q}_{c5}^{\mathrm{T}} \ \mathbf{Q}_{s5}^{\mathrm{T}}\} \tag{32}$$

The solution of Equation (31) is performed using an arc-length strategy. In particular, the continuation is conducted with respect to the frequency parameter $\Omega = \lambda \omega_{11}$, where $\omega_{11}$ is the linear fundamental frequency, while $\lambda$ is the control parameter. Owing to the additional unknown introduced via $\Omega$, one constraint equation is needed. Among the different approaches available in the literature, the Crisfield equation is considered [42]:

$$g(\mathbf{x}, \lambda) \equiv \Delta \mathbf{x}^{\mathrm{T}} \Delta \mathbf{x} + \Delta \lambda^2 \omega_{11}^2 - \Delta l^2 = 0 \tag{33}$$

Incremental quantities are denoted with $\Delta$ and refer to the previously converged solution, i.e., $\Delta \mathbf{x} = \mathbf{x} - \mathbf{x}^{(k)}$ and $\Delta \lambda = \lambda - \lambda^{(k)}$; the scalar $\Delta l$ is the fixed arc-length increment.

### 3.4. Single-Mode Solution

The two methods illustrated earlier allow the solution to be calculated with relative ease, the computational time being of the order of tens of seconds. In many design situations, further time reduction can be desirable, especially when thousands of analyses are necessary for preliminary optimization, parametric studies or analysis of the sensitivities to the design variables. These considerations appear particularly relevant when the target of the design process is a variable stiffness configuration: under these circumstances, the number of design variables is generally much larger with respect to equivalent straight-fiber designs, so the need to perform several analysis runs is even more clear. In this context, the availability of fast design strategies is of paramount importance as any reduction in the computational time has a multiplicative effect on the total time of the procedure. Aiming at providing an answer to this need, a single-mode approach was developed. Hereinafter, it will be denoted as Harmonic Balance Single-mode (HBS). This solution inherently embeds a larger degree of approximation—it is indeed a simplified version of Equation (24)–but allows trends to be appropriately captured in fractions of the second. The single-mode version of Equation (24) reads:

$$\ddot{q} + 2\zeta\omega\dot{q} + \omega^2 q + \beta q^3 = f \cos(\Omega t) \tag{34}$$

The solution of Equation (34) can be found with relative ease by assuming that the contribution of super-harmonics is negligible. The Fourier expansion of Equation (30) is then truncated at the first term, i.e.,:

$$q(t) = Q_{c1} \cos(\Omega t) + Q_{s1} \sin(\Omega t) \tag{35}$$

Based on the approximation of Equation (35), the cubic term appearing in Equation (34) is expressed as:

$$\begin{aligned} q^3 &= 3Q_{c1}Q_{s1}^2 \cos(\Omega t) + 3Q_{c1}^2 Q_{s1} \sin(\Omega t) + \left( Q_{c1}^3 - 3Q_{c1}Q_{s1}^2 \right) \cos^3(\Omega t) + \left( Q_{s1}^3 - 3Q_{c1}^2 Q_{s1} \right) \sin^3(\Omega t) \\ &\approx \frac{3}{4} Q_{c1} \left( Q_{c1}^2 + Q_{s1}^2 \right) \cos(\Omega t) + \frac{3}{4} Q_{s1} \left( Q_{c1}^2 + Q_{s1}^2 \right) \sin(\Omega t) \end{aligned} \tag{36}$$

where the expression is approximated after dropping the superharmonic terms obtained by elaboration of $\cos^3(\Omega t)$ and $\sin^3(\Omega t)$. The final set of solving equations is obtained upon substitution of Equation (36) into Equation (34), followed by the balancing of corresponding harmonic terms. Two nonlinear algebraic equations are obtained in the form:

$$
\begin{cases}
-\Omega^2 Q_{c1} + 2\zeta\omega\Omega Q_{s1} + \omega^2 Q_{c1} + \dfrac{3}{4}\beta Q_{c1}\left(Q_{c1}^2 + Q_{s1}^2\right) = f \\
-\Omega^2 Q_{s1} - 2\zeta\omega\Omega Q_{c1} + \omega^2 Q_{s1} + \dfrac{3}{4}\beta Q_{s1}\left(Q_{c1}^2 + Q_{s1}^2\right) = 0
\end{cases}
\tag{37}
$$

where the unknowns are given by the amplitudes of the sine and cosine parts of the solution postulated in Equation (35). The two equations above can be merged into one single scalar equation if the modal amplitude $|q| = \sqrt{Q_{c1}^2 + Q_{s1}^2}$ is introduced. Accordingly, by squaring the two expressions of Equation (37), one obtains:

$$
\left[\left(-\omega^2 + \Omega^2 - \frac{3}{4}\beta|q|^2\right)^2 + (2\zeta\omega\Omega)^2\right]|q|^2 = f^2
\tag{38}
$$

This simple scalar equation provides a closed-form solution for the amplitude-versus-frequency relation. Given a frequency value $\Omega$, the corresponding amplitude value $|q|$ is readily found. Note that Equation (38) is a sixth order polynomial with even powers of $|q|$ only. It is then possible to reduce it to a third-order polynomial equation from which a closed-form solution is found.

## 4. Results

The potential of the proposed semi-analytical methods is discussed in this section. The first part deals with the validation of the methods against reference results from the literature. In the second part, focus is given to the comparison between different solution strategies, i.e., DQ, HB, and closed-form solution. Specifically, the comparison is illustrated in terms of accuracy and computational time. In the third part, the reduced time of the closed-form approach is exploited to illustrate its potential use in the context of a parametric study. The results are reported in the form of design charts, which are believed to be of interest to gather a quick understanding of the structural response of variable stiffness designs.

### 4.1. Validation

The proposed approach relies upon two peculiar features, regarding the combined possibility of handling variable stiffness configurations in combination with TSDT. Accordingly, the validation requires these two aspects to be verified, independently or in combination. Exact reference solutions are not available in the literature for TSDT plates with non uniform stiffness distribution. For this reason, the validation phase is split into two parts. Firstly, the comparison is illustrated for composite plates with uniform stiffness, i.e., straight-fibers: indeed, results are available in exact form for this case [29] and represent an excellent reference for validation purposes. In a second step, the comparison is conducted for VS plates, where benchmarks are available in the form of finite element simulations [21,43].

#### 4.1.1. Linear Vibrations

This section deals with the validation of the formulation in terms of linear free vibrations. The elastic properties of the materials considered herein are summarized in Table 1; the density is assumed to be $1540\,\text{kg/mm}^3$, unless otherwise specified. Different geometries are considered in the section and specified case by case.

**Table 1.** Material elastic properties.

|  | $E_1$ (MPa) | $E_2$ (MPa) | $G_{12}$, $G_{13}$ (MPa) | $G_{23}$ (MPa) | $\nu_{12}$ |
|---|---|---|---|---|---|
| M1 | $f(E_1/E_2)$ | 1.00 | 0.60 | 0.50 | 0.25 |
| M2 | 120,500 | 9630 | 3580 | 3580 | 0.32 |
| M3 | 173,000 | 7200 | 3760 | 3760 | 0.29 |
| M4 | 131,700 | 9860 | 4210 | 4210 | 0.28 |

A square, simply-supported plate is investigated. Two different length-to-thickness ratios $a/h$ are considered in order to showcase the effect of transverse shearing deformability. The material corresponds to M1 of Table 1, where the orthotropy ratio $E_1/E_2$ is varied between 10 and 40, and the cross-ply lay-up is $[0/90]_s$. The results are presented in Table 2 in terms of the nondimensional first linear frequency $\overline{\omega}_1 = \omega \dfrac{a^2}{h} \sqrt{\dfrac{\rho}{E_2}}$. The computations are performed by considering a different number of trial functions—and so different number of degrees of freedom $n_{dof}$—and are compared against the Navier solution [29].

**Table 2.** Frequency parameter $\overline{\omega}_1$ for straight-fiber plates: comparison with [29].

| $E_1/E_2$ | $a/h$ | $n_{dof} = 48$ | $n_{dof} = 108$ | $n_{dof} = 363$ | **Exact [29]** |
|---|---|---|---|---|---|
| 10 | 5 | 8.2731 | 8.2718 | 8.2718 | 8.272 |
|  | 10 | 9.8430 | 9.8410 | 9.8410 | 9.841 |
| 20 | 5 | 9.5276 | 9.5263 | 9.5263 | 9.526 |
|  | 10 | 12.2205 | 12.2180 | 12.2180 | 12.218 |
| 40 | 5 | 10.7885 | 10.7873 | 10.7873 | 10.787 |
|  | 10 | 15.1100 | 15.1073 | 15.1073 | 15.107 |

As seen, the exact solution of the problem is matched up to the third digit with as few as 48 degrees of freedom. The proposed expansion is indeed characterized by excellent convergence properties and, in general, relatively few functions suffice to achieve accurate results. When the number of trial functions is increased, the solution converges towards the exact solution for the remaining digits. One should note that convergence is achieved from above, meaning that an increase in the number of trial functions is always associated with a reduction of the natural frequencies. However, this conclusion cannot be extended to VS configurations. It should be noted that the results of Table 2 are restricted to the first natural frequency and, in general, more trial functions could be needed if high-order modes are of interest.

The second example deals with a VS configuration and aims at illustrating the ability of the method to handle correctly the case on non-uniform stiffness distribution. The planar dimensions are taken as $1000 \times 1000$ mm$^2$. The thickness is equal to 10 mm, corresponding to a ratio $a/h$ of 100. The plate is made of the material M3 and the symmetric stacking sequence $[\langle 0|45 \rangle, \langle -45|-60 \rangle, \langle 0|45 \rangle]$ is assumed. The boundary conditions correspond to simply-supported and clamped edges. The results are reposted in Table 3, where the comparison is presented against the results using the p- and h-version of FEM [21,43], respectively.

To emphasize the efficiency of the proposed formulation, the results are reported by considering two expansions: in one case, the total number of dofs is taken equal to 506, which is close to the problem size considered in [21,43]; in the second case, the number of dofs is 156, corresponding to the smallest number of functions to guarantee converged results.

**Table 3.** Natural frequencies (rad/s) for VS plates: comparison with [21,43].

| Mode | Ref. [21] $n_{dof} = 500$ | | Ref. [43] $n_{dof} = 500$ | | Ritz $n_{dof} = 506$ | | Ritz $n_{dof} = 156$ | |
|---|---|---|---|---|---|---|---|---|
| | **SSSS** | **CCCC** | **SSSS** | **CCCC** | **SSSS** | **CCCC** | **SSSS** | **CCCC** |
| 1 | 355.41 | 567.56 | 358.49 | 579.40 | 357.30 | 575.62 | 360.70 | 578.08 |
| 2 | 600.50 | 831.39 | 589.90 | 821.53 | 589.12 | 818.39 | 595.32 | 821.76 |
| 3 | 986.65 | 1253.18 | 960.36 | 1225.79 | 960.46 | 1222.74 | 980.81 | 1254.50 |
| 4 | 1027.55 | 1448.46 | 1075.21 | 1493.76 | 1073.03 | 1479.45 | 1086.21 | 1500.97 |
| 5 | 1309.92 | 1719.96 | 1327.88 | 1726.96 | 1322.84 | 1713.21 | 1350.30 | 1764.76 |
| 6 | 1506.90 | 1818.98 | 1474.67 | 1775.16 | 1466.67 | 1771.14 | 1690.45 | 1908.24 |
| 7 | 1743.33 | 2175.80 | 1726.71 | 2135.76 | 1718.08 | 2121.56 | 2265.94 | 2220.54 |
| 8 | 2106.31 | 2505.73 | 2137.13 | 2443.53 | 2085.50 | 2437.05 | 2324.50 | 2772.34 |
| 9 | 2171.03 | 2750.46 | 2262.35 | 2706.78 | 2227.60 | 2690.78 | 2591.59 | 2947.36 |

Close agreement can be noted in Table 3 between the Ritz solutions and the FE ones. Using the proposed Ritz approach, a drastic savings can be achieved in terms of problem size. The number of dofs is approximately 1/3 the finite elements to achieve similar accuracy.

The shapes of the first four modes are presented in Figure 3. For the simply-supported case, the modal shapes are available even in [21]. Close agreement can be noted with the reference results both in terms of number of halfwaves as well as the skewness induced by the plate elastic couplings. The modal shapes for the clamped plate are not provided in the referenced works, but are reported here for the sake of completeness. These modes exhibit similar patterns with respect to the ones of the SSSS plates, apart from the boundary effects which lead to vanishing rotations at the four edges.

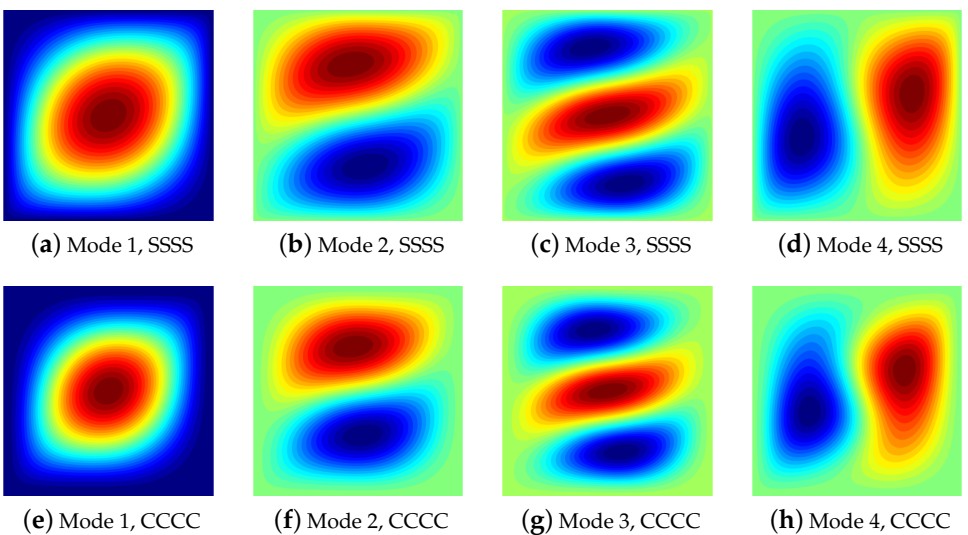

(**a**) Mode 1, SSSS     (**b**) Mode 2, SSSS     (**c**) Mode 3, SSSS     (**d**) Mode 4, SSSS

(**e**) Mode 1, CCCC     (**f**) Mode 2, CCCC     (**g**) Mode 3, CCCC     (**h**) Mode 4, CCCC

**Figure 3.** Vibration shapes of plates with layup $[\langle 0|45\rangle, \langle-45|-60\rangle, \langle 0|45\rangle]$: (**a**–**d**) simply-supported; (**e**–**h**) clamped.

4.1.2. Nonlinear Vibrations

A first comparison in terms of nonlinear response is conducted by referring to the test cases proposed by Ribeiro [20]. A rectangular plate of dimensions $320 \times 480$ mm$^2$ is considered. The material is M2 and the layup is $[\langle\theta+45|90\rangle, \langle-\theta|-45\rangle, \langle\theta|45\rangle, \langle\theta-45|0\rangle]_s$, with $\theta$ equal to 90. The total thickness is 1 mm. The plate is constrained at the four edges with fully clamped and immovable boundary conditions, see Figure 2. The frequency–response curve obtained with the DQ method is reported in Figure 4 by considering different load intensities $f$, ranging from 0.2 to 1.2 N, where the external load is applied in

the form of a concentrated load at the plate center. The modal basis is composed by the the first, third and seventh linear eigenmodes, based on a preliminary assessment. The time parameters for the DQ method are $N_e = 42$ and $N = 40$. The nondimensional frequency $\mathcal{R}$ is defined as $\mathcal{R} = \dfrac{\Omega}{\omega_1}$, with $\omega_1$ representing the first linear fundamental frequency. The peaks of the frequency–response plots provide the backbone curve to be compared with the one derived by Ribeiro [20] using FSDT theory, a p-FEM approach for the spatial approximation and the Harmonic Balance Method for handling the time dependency.

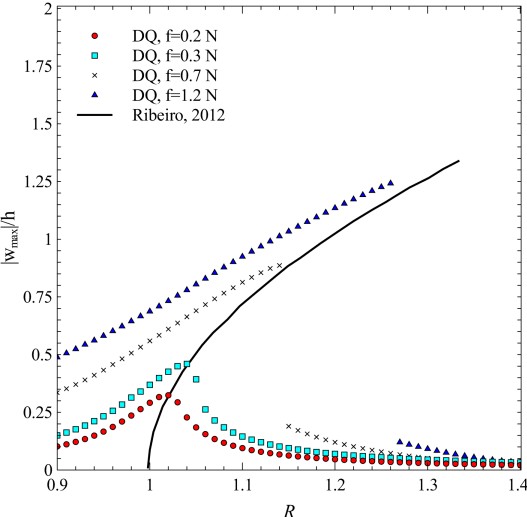

**Figure 4.** DQ method–comparison with [20].

The results demonstrate close agreement with the reference solution. Slight discrepancies occur and can be motivated by the combined effects of different kinematic theories and different approximations for the spatial and temporal dependencies. In general, the DQ approach in its standard implementation does not allow the unstable part of the curve to be captured. For this reason, sudden jumps can be noted in the Ritz solution from the high- to the low-amplitude range observed for $f \geq 0.4\,\text{N}$.

The same test case is analyzed to illustrate the comparison with the formulation based on the HB method. Based on a preliminary assessment, the first and the third harmonics are retained in the Fourier expansion. The results are summarized in Figure 5. For consistency, the plot is presented with the same format reported for the DQ method.

Close agreement can be noted against the reference solution even in this case. The peaks of the Ritz predictions are even closer to the solutions reported by Ribeiro. One reason for this improved correspondence lies in the same approximation for the temporal part employed here and in [20]. As a matter of fact, both approaches rely on the HB method. Owing to the arc-length strategy illustrated earlier, the unstable branch is easily captured by the proposed HB method, and no jumps occur at the larger load levels. The comparison with the backbone curve of Figure 5 is straightforward. One can note the milder hardening effect exhibited by the present solution: this is consistent with the treatment of the in-plane boundary conditions, which are imposed in an average sense due to the adoption of a mixed approach rather than a displacement-based one.

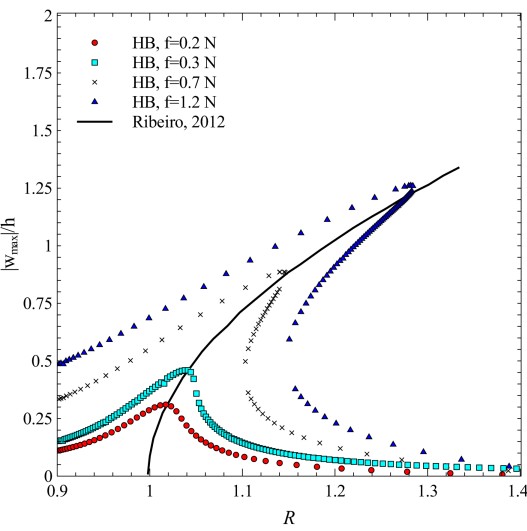

**Figure 5.** HB method–comparison with [20].

A second example regards the nonlinear vibration response of the variable stiffness plates studied in [22]. The dimensions are $a = b = 500$ mm, while the total thickness is 5 mm. The plate is layered with four plies made of material M3 and stacking sequence $[\mp\langle 40|\alpha\rangle]_s$, where $\alpha$ varies between 10 and 90. The panel is fully clamped with immovable in-plane restraints. The solution is obtained using the HBS method to illustrate the potential of this fast strategy in estimating the nonlinear vibration frequencies. The displacements are expanded with 10 terms along both directions, then the first mode is retained following the procedure illustrated in Section 3.4. The results are then obtained by solution of Equation (38).

A summary of the numerical predictions is available in Table 4, where the nonlinear frequency parameter $\hat{\omega}_{nl} = \omega_{nl}a\sqrt{\frac{\rho}{E_2}}$ is reported for different values of the maximum nondimensional deflection and different layups.

**Table 4.** Nonlinear frequency parameters $\hat{\omega}_{nl} = \omega_{nl}a\sqrt{\frac{\rho}{E_2}}$: comparison with [22].

| $|w_{\mathbf{max}}|/h =$ | 0.2 | | 0.6 | | 1.0 | |
|---|---|---|---|---|---|---|
| | **HBS** | **Ref. [22]** | **HBS** | **Ref. [22]** | **HBS** | **Ref. [22]** |
| $[\mp\langle 40|10\rangle]_s$ | 0.316 | 0.325 | 0.335 | 0.345 | 0.370 | 0.383 |
| $[\mp\langle 40|20\rangle]_s$ | 0.310 | 0.318 | 0.328 | 0.338 | 0.362 | 0.375 |
| $[\mp\langle 40|30\rangle]_s$ | 0.302 | 0.308 | 0.319 | 0.328 | 0.351 | 0.364 |
| $[\mp\langle 40|40\rangle]_s$ | 0.292 | 0.297 | 0.308 | 0.316 | 0.338 | 0.350 |
| $[\mp\langle 40|50\rangle]_s$ | 0.282 | 0.287 | 0.297 | 0.304 | 0.325 | 0.335 |
| $[\mp\langle 40|60\rangle]_s$ | 0.273 | 0.278 | 0.286 | 0.294 | 0.311 | 0.322 |
| $[\mp\langle 40|70\rangle]_s$ | 0.267 | 0.272 | 0.279 | 0.287 | 0.300 | 0.313 |
| $[\mp\langle 40|80\rangle]_s$ | 0.264 | 0.269 | 0.275 | 0.283 | 0.296 | 0.309 |

All the structures display a hardening-type response—consistently with the reference results—with increasing values of the frequencies for increasing deflections. The comparison against the finite element solution proposed by Houmat [22] reveals a good degree of accuracy: the differences in the predicted frequency are always well below 5%. For a given nondimensional amplitude, the reference frequencies are always larger than the present ones. The same behavior has been observed in a previous work [25] and is mainly motivated by the different handling of the in-plane boundary conditions. The reference work relies upon a displacement-based approach, where in-plane constraints are enforced in a strong-form manner. Within the mixed formulation adopted here, in-plane conditions

are handled in a weak-form sense, leading to some additional compliance due to the local violation of these conditions.

### 4.2. Comparison between Methods

In this section, an exemplary test case is presented to illustrate the comparison between the different strategies reported in the paper. In particular, the comparison involves the DQ and the HB methods, this latter implemented in the multi-mode and simplified, single-mode formulations.

The plate under investigation is the same considered in Section 4.1.2, the only difference relying the stacking sequence, which is now $[\langle \theta + 45|90 \rangle, \langle -\theta| -45 \rangle, \langle \theta|45 \rangle, \langle \theta - 45|0 \rangle]_s$, with $\theta$ equal to 0. Clamped conditions with immovable edges are considered.

The linear mode shapes of the plate are summarized in Figure 6, where the first eight eigenvectors are summarized. These shapes tend to be relatively complex due to the stiffness non-uniformity.

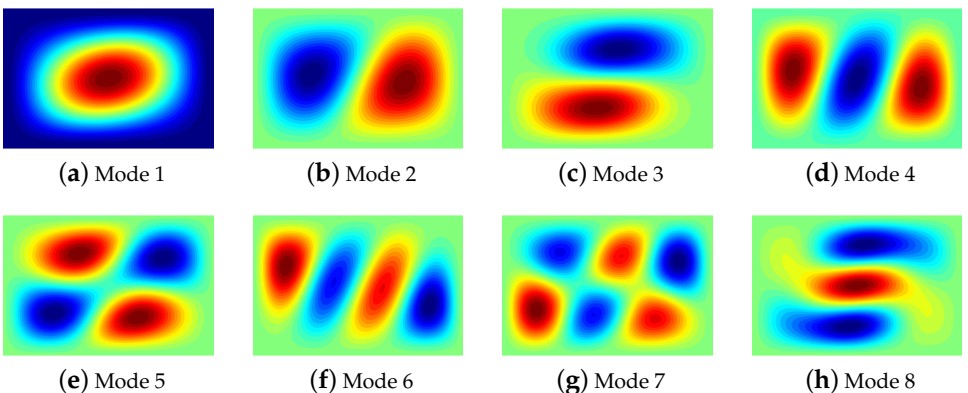

| (**a**) Mode 1 | (**b**) Mode 2 | (**c**) Mode 3 | (**d**) Mode 4 |
|:---:|:---:|:---:|:---:|
| (**e**) Mode 5 | (**f**) Mode 6 | (**g**) Mode 7 | (**h**) Mode 8 |

**Figure 6.** Vibration shapes of plates with layup $[\langle 45|90 \rangle, \langle 0| -45 \rangle, \langle 0|45 \rangle, \langle -45|0 \rangle]_s$.

A preliminary assessment was conducted to identify the effect of adopting different bases for studying the nonlinear response. In particular, it was found that the frequency–response curve is mainly affected by the modes 1, 4 and 8, which are then retained in the DQ and HB approaches for comparison purposes against the single-mode one. Further expansion of the basis does not provide any noticeable improvements. The results are presented in Figure 7, where the load is applied as a concentrated force in the middle and the magnitudes are taken equal to 0.5 and 1.0 N.

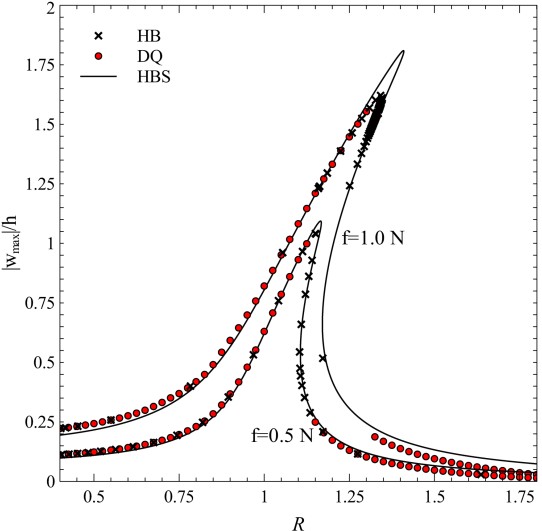

**Figure 7.** Comparison between different methods.

The maximum nondimensional displacements $w_{\max}/h$ observed in Figure 7 are 1.1 and 1.8, approximately. In the first case, the degree of nonlinearity is mild, while the geometric nonlinear effects are much more relevant when the force is increased to 1.0 N.

As seen from Figure 7, the three methods lead to similar results when the degree of nonlinearity is relatively small. On the contrary, slight discrepancies can be noted in the presence of a larger degree of geometric nonlinearity. As seen, the single-mode HB solution tends to be stiffer with respect to the multi-mode ones in the small-amplitude regime. Here, the effect of membrane stretching is small, and the approximations due to the integral representation of the in-plane boundary conditions are negligible. The single-mode solution is found to be more compliant when the amplitude of vibration increases—see the central portion of the plot in Figure 7—as far as the membrane effects are more pronounced, and the approximations of in-plane conditions are more and more relevant.

It is interesting to provide a quantitative comparison between the three strategies outlined earlier in terms of nondimensional deflections $w_{\max}/h$ and time for the analysis. The results are summarized in Table 5, where computations are carried out with a standard laptop with i7 processor and 16 GB of RAM.

**Table 5.** Comparison between different methods.

|  |  | $|w_{\mathbf{max}}|/h$ | Time (s) |
|---|---|---|---|
| DQ | $f = 0.5$ N | 1.0427 | 4.3 |
| HB |  | 1.0451 | 8.5 |
| HBS |  | 1.0875 | $\ll 0.1$ |
| DQ | $f = 1.0$ N | 1.6502 | 9.5 |
| HB |  | 1.6668 | 7.6 |
| HBS |  | 1.8080 | $\ll 0.1$ |

The HB and the DQ approaches lead to similar results. This is somewhat expected, as they differ by the handling of the time dependency, but no major sources of approximation are introduced when turning from one to another. The time for the analysis is of the order of 10 s, which is attractive for analysis purposes. In the context of preliminary design studies, any further time reduction is amplified by the number of runs to be performed. In this regard, the single-mode HB approach offers an interesting trade-off between accuracy and CPU cost. In this case, the predictions are, in general, less accurate with respect to the multi-modal ones. However, drastic reductions of the cpu times can be achieved, with single runs requiring much less than a second. Therefore, the trade-off between accuracy and computational cost is believed to be particularly interesting. The single-mode strategy is the ideal candidate for performing parametric studies and capturing trends with improved computational efficiency.

### 4.3. Parametric Studies

Based on the results of the previous section, a study is presented to illustrate the advantages of a fast and efficient approach to perform a preliminary assessment. A VS laminate is considered whose dimensions are $420 \times 300$ mm$^2$. The total thickness is 4 mm. The material properties of M4 are considered, and the density is equal to 1600 kg/mm$^3$. Simply-supported boundary conditions are considered at the four edges along with in-plane immovable conditions. A concentrated force $f$ of 100 N is applied at the center of the plate.

The layup $[\langle T_0|45\rangle, 90\langle T_0|45\rangle]_s$ is considered. The goal of the assessment is investigating the influence of the angle $T_0$ on the plate nonlinear frequency–response behavior.

The results are summarized in Figure 8, where the curves are traced with $T_0$ ranging from 0 to 90 with steps of 45. The configuration associated with $T_0 = 45$ is a straight-fiber laminate and is reported in the plot for comparison purposes.

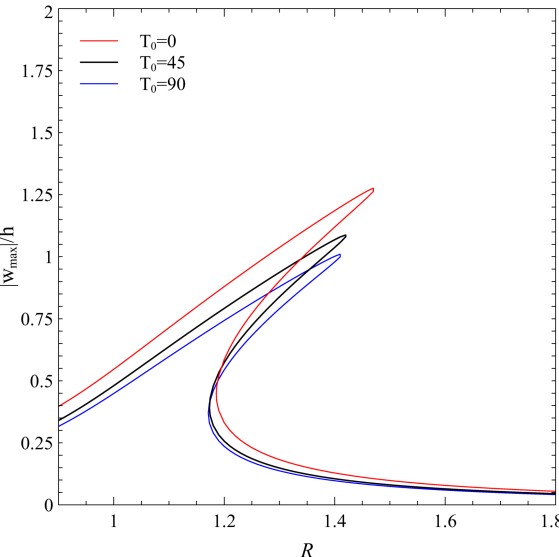

**Figure 8.** VS laminate $[<T_0|45>, 90<T_0|45>]_s$: effect of angle $T_0$.

As shown in Figure 8, the maximum deflection is achieved at the same nondimensional frequency $\mathcal{R}$ for all the configurations; however, the amplitude of the deflection varies dependently on the layup. Specifically, larger values of $T_0$ tend to reduce the deflection peak and increase the hardening effect. When comparing the VS solutions to the straight-fiber ones, one can observe that larger peak reductions can be achieved thanks to steering.

On the basis of the results of Figure 8, it is concluded that the orientation angle $T_0$ can be tuned to meet the design requirements, leading to a much larger freedom in shaping the laminate response as compared to a straight-fiber design. In this study, the investigation is restricted to the frequency–response curve, but other metrics will have to be accounted for in real design scenarios.

A second study is presented by assuming a layup $[\mp\langle T_0|T_1\rangle]_{2s}$, where the design parameters to be explored are given by the angles $T_0$ and $T_1$. For simplicity, manufacturing constraints are not considered at this stage. Two structural responses are investigated, corresponding to the maximum nondimensional amplitude $w_{max}/h$ and the degree of hardening $\theta_H$. This latter is intended as the slope, expressed in degrees, of the linearized force response curve in the interval between $\mathcal{R}$ equal to 1 and $\mathcal{R}^*$, where $\mathcal{R}^*$ is the nondimensional frequency corresponding to the amplitude peak. The lower the values of $\theta_H$, the higher the degree of hardening exhibited by the laminate.

The contour plots of Figure 9 summarize the structural behaviour for any combination of $T_0$ and $T_1$. Straight-fiber configurations lie on the diagonal of the plots, so comparisons with VS configurations can be easily drawn.

As shown, the plate response can be tuned with relative ease to match the requirements regarding the two responses considered here. For instance, the maximum deflection can be reduced through a VS design. Similarly, the degree of hardening can be increased or decreased through proper selection of $T_0$ and/or $T_1$, depending on the design needs and the potential requirements associated with other structural responses.

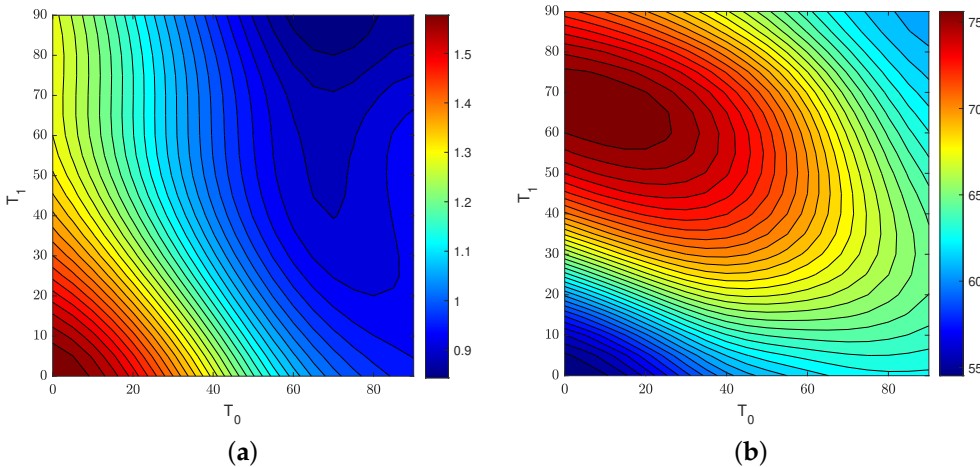

**Figure 9.** VS laminate $[\mp\langle T_0|T_1\rangle]_{2s}$: (**a**) maximum deflection $w_{\max}/h$; (**b**) slope $\theta_H$.

## 5. Conclusions

This work illustrates a new mixed formulation for the analysis of laminated plates with non uniform stiffness properties. The effect of transverse shear deformability is included owing to TSDT. The formulation is developed within a variational context, and appears to be the first attempt to combine a mixed strategy with TSDT. This is of particular interest when the application of direct solution methods is sought, as it is the case for plates with spatially varying elastic properties. The framework derived here is not restricted to nonlinear dynamics, but could be equivalently employed for problems in statics where geometrical nonlinearity is of concern.

The main advantage of the approach lies in the possibility of reducing the total number of theory-related degrees of freedom from five to four. This feature, along with the good convergence properties of the Ritz method, allows semi-analytical procedures to be developed with improved efficiency.

In this work, the application of the proposed variational principle is presented to the nonlinear dynamic response of VS plates, with a focus on nonlinear free vibrations. The main result regards the possibility of successfully combining different techniques for handling the spatial and the temporal dependency within the proposed variational framework. Differential Quadrature and Harmonic Balance methods were proposed for this purpose. The accuracy and the efficiency of the procedures were illustrated by comparison against reference results from the literature.

The time for the analysis is very attractive, with cpu times of the order of 10 s for the multi-modal solutions and fractions of seconds for the single-mode approach. The former allow improved accuracy and are particularly suitable for analysis purposes, the latter is characterized by reduced accuracy, but is of particular interest when preliminary assessments are of concern. In particular, the single-mode solution proved to be a powerful tool for investigating the effect of the many design parameters typical of VS configurations.

General guidelines regarding the optimal fiber path can be hardly provided given the dependency on the response of interest, as well as the plate geometry and its boundary conditions. However, as demonstrated for some test cases, the additional tailoring chances offered by a VS design can be effectively exploited.

**Author Contributions:** The two authors equally contributed to the present paper. All authors have read and agreed to the published version of the manuscript.

**Funding:** Riccardo Vescovini would like to thank the Ministero dell'Istruzione, dell'Università della Ricerca for funding this research under the PRIN 2017 program.

**Data Availability Statement:** Not applicable.

**Conflicts of Interest:** The authors declare no conflict of interest.

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
