# Peer review of "Simplified Approach to Nonlinear Vibration Analysis of Variable Stiffness Plates"

_jcs, doi:10.3390/jcs7010030_

Round 1
Reviewer 1 Report
The paper presents an interesting formulation for the analysis of the nonlinear vibrations of variable stiffness laminated plates. The theoretical framework is based on the TSDT, whereas the solution is obtained by means of the Ritz approximation (in space) and Differential Quadrature together with Harmonic Balance (in time). The authors present several numerical applications to show the advantages and the accuracy of the proposed methodology.
In general, the paper is well-written. Both theoretical and numerical sections are clear and focused on the more innovative aspects introduced by the research. The topic is well-introduced and the proper references have been mentioned. The combination of TSDT and a mixed approach is interesting as well.
In the following, some minor suggestions and questions to improve the quality of the paper (which is suitable for publication in the Journal):
1. With reference to Eq.(1), please cite the proper reference if the expression is taken from other works.
2. Two different abbreviations are used for the Differential Quadrature, which are DQ and DQM. Please, uniform the notation by employing DQ everywhere in the paper.
Overall, the quality of the paper is high. Therefore, it can be published with minor revision.
Reviewer 2 Report
Review of the manuscript by Jorge Andérez González and Riccardo Vescovini
Simplified Approach to Nonlinear Vibration Analysis of Variable Stiffness Plates
Submitted to J. Compos. Sci. (2100067)
GENERAL COMMENTS
The paper deals with the nonlinear vibration behaviors of variable stiffness plates. The mathematical formulations are presented, and the approximate solutions are obtained by using the differential quadrature and harmonic balance methods. Some parametric studies are presented to show the nonlinear dynamics of the VS plates. However, the motivation is not illustrated very clearly, and the creativity of the manuscript is not significant. Some parametric studies are elementary, and more illustrations and investigations should be added. According to the high-quality standards of the J. Compos. Sci., the paper CAN NOT be considered for publication in the present version. Some major and minor comments are summarized as follows.
MAJOR/MINOR COMMENTS
u The INTRODUCTION is essentially a disordered list of concise (sometimes vague and imprecise) statements about what other authors did in the past in the field. The mere sum of these statements is far from a coherent analysis of the current state of the art and certainly does not suffice to support the paper’s motivations. The authors must relate the referenced works to each other by highlighting the advancements and significant theoretical/applied results of each piece of research. The motivation of the manuscript should be illustrated more clearly and concisely. The present introduction is poor, and more illustrations and some relative references should be added.
u The derivation of the mathematical model should be illustrated more clearly. The readability of this section is reduced.
u There are many different types of asymptotic methods, so why the authors choose these methods in the study.
u The presented HBM is very classical, so the calculation procedure should be reduced.
u Based on the Galerkin method, the partial differential equations have been reduced to ordinary ones. However, only the single mode shape has been considered in the discrete equations (Eq. 43). In fact, there are some significant quantitative and qualitative differences between the single and multiple mode discretizations. More illustrations and studies on the mode discretizations should be added. It is a significant factor in the nonlinear dynamics.
u As to Eq. (43), there is just cubic nonlinearity. It is very easy to solve this type of equation.
u In the linear analysis, only the natural frequencies are presented and compared. More illustrations on the mode shapes should be added here. It is also a significant problem.
u Only some frequency response curves are presented.
u Instead of frequency response curves, there are many ways to demonstrate the nonlinear system's vibration behaviors, e.g., force-response amplitude curves, time-history curves, phase diagrams, Poincare sections, frequency spectrums, Lyapunov exponents, etc. More figures and investigations could be provided to exhibit the vibration behaviors of the VS plates.
u A very important factor is neglected completely in the nonlinear analysis: the stability. More illustrations on the stability should be added in the nonlinear analyses.
u Some parametric analyses have been discussed, but many similar frequency-response curves have been illustrated. Appropriate simplifications are necessary in the revised manuscript, and only the significant and interesting phenomena should be provided. It is suggested that the authors should highlight some significant research results only.
u Some figures are very elementary and poor. e.g., Fig. 7. It seems that these two figures are discontinuous around the jump point (the saddle-node bifurcation).
u Some relative references on the plates are not cited, e.g., Saetta and Rega et al.
u It seems that the conclusion is feeble and requires to be diffusively strengthened. Some conclusions are not new at all.
Round 2
Reviewer 1 Report
The paper is now suitable for publication in the current form. All the comments and observations provided by the reviewers have been included in the revised version of the manuscript. No further changes are required.
Reviewer 2 Report
The authors have satisfyingly answered to almost all the points raised in my initial review. My opinion is that the paper has been improved, and according to the high quality of J. Compos. Sci, the paper can be acceptable after the grammar is double checked.